# Insulin Downregulated the Infection of Uropathogenic *Escherichia coli* (UPEC) in Bladder Cells in a High-Glucose Environment through JAK/STAT Signaling Pathway

**DOI:** 10.3390/microorganisms9122421

**Published:** 2021-11-24

**Authors:** Chen-Hsun Ho, Shih-Ping Liu, Chia-Kwung Fan, Kai-Yi Tzou, Chia-Chang Wu, Po-Ching Cheng

**Affiliations:** 1Division of Urology, Department of Surgery, Shin Kong Wu Ho-Su Memorial Hospital, Taipei 11101, Taiwan; chho.uro@gmail.com; 2School of Medicine, College of Medicine, Fu Jen Catholic University, New Taipei City 24205, Taiwan; 3Department of Urology, National Taiwan University Hospital and College of Medicine, Taipei 10002, Taiwan; spliu@ntuh.gov.tw; 4Department of Molecular Parasitology and Tropical Diseases, School of Medicine, College of Medicine, Taipei Medical University, Taipei 11031, Taiwan; tedfan@tmu.edu.tw; 5Center for International Tropical Medicine, School of Medicine, College of Medicine, Taipei Medical University, Taipei 11031, Taiwan; 6Department of Urology, School of Medicine, College of Medicine, Taipei Medical University, Taipei 11031, Taiwan; 11579@s.tmu.edu.tw; 7Department of Urology, Shuang Ho Hospital, Taipei Medical University, New Taipei City 23561, Taiwan; 8TMU Research Center of Urology and Kidney (TMU−RCUK), Taipei Medical University, Taipei 11031, Taiwan

**Keywords:** bladder cells, insulin, JAK/STAT, urinary tract pathogenic *Escherichia coli*

## Abstract

Diabetic individuals have a higher incidence of urinary tract infection (UTI) than non-diabetic individuals, and also require longer treatment. We evaluated the effects of insulin pretreatment on the regulation of JAK/STAT transduction pathways in UPEC-infected bladder cells in a high-glucose environment. A bladder cell model with GFP-UPEC and fluorescent-labeled TLR4, STAT1, STAT3, and insulin receptor antibodies, was used to evaluate the relationship between insulin receptor signaling, TLR-4-mediated, and JAK/STAT-dependent pathways. Pretreatment with 20 and 40 µg/mL insulin for 24 h significantly and dose-dependently reduced UPEC infection in SV-HUC-1 cells. Additionally, the expression levels of STAT1 and STAT3 were downregulated in a dose-dependent manner. However, insulin receptor (IR) expression was not affected by insulin pretreatment. Our results showed that insulin-mediated reduction of UPEC infection in a high-glucose environment was not only due to the downregulation of JAK1/2 and phosphorylated STAT-1/3, but also because of the decreased expression of TLR-4 proteins and pro-inflammatory IL-6. Here, we demonstrated that insulin reduced not only UPEC infection in bladder epithelial cells, but also inhibited the JAK/STAT transduction pathway during infection in a high-glucose environment. This study provides evidence to support the use of insulin in the treatment of UPEC infection in patients with type 2 diabetes (T2D).

## 1. Introduction

UTI is an infection in any part of the urinary system, including the kidneys, ureters, bladder and urethra, especially the lower urinary tract, the bladder and the urethra. Some studies point out that 1/3 of women will develop UTI during their lifetime [1,2]. Most cases of cystitis are caused by UPEC [3]. According to the literature, 76.7% of the bacteria in female patients with urinary tract infections are *E. coli* [4]. Bacterial cystitis, also known as a kind of UTI, usually occurs when bacteria from outside enter the urethra and begin to proliferate, often accompanied by frequent urination, urgency, burning or pain during urination, hematuria, and other symptoms. The onset is usually very rapid, causing severe discomfort and even spreading to the kidneys to form acute pyelonephritis [5]. A couple of studies have revealed bacterial cystitis can cause infections in young female after sexual intercourse, but women who do not often engage in sexual activities may also be at risk of UTI because the anatomical architecture of female urogenital tract makes it prone to bacterial infection of cystitis [6].

Patients suffering from diabetes mellitus (DM) and UTI at the same time have a more complicated situation that requires longer treatment than patients without diabetes [7]. Moreover, in these patients, the chances of complications such as acute nephritis, renal abscess, emphysematous pyelonephritis, and emphysematous cystitis are relatively high [8,9]. In addition, long-term high blood sugar levels can also weaken the immune system [2,10]. High concentrations of urine sugar also provide more nutrients to pathogenic bacteria [11,12]. In a previous study, increased adherence of *E. coli* pili to uroepithelial cells was observed in diabetic patients compared to non-diabetic patients, and it was positively correlated with the level of glycosylated hemoglobin, causing recurrent UTIs [13]. The pili of *E. coli* are glycoproteins, and the uroepithelial cells in patients with diabetes possess glycoprotein receptors, which favors the adhesion and increases the chances of infection [7].

Toll-like receptors are greatly increased in various inflammatory diseases and diabetes. Recent studies have mentioned that the Asp299Gly polytype in the TLR4 gene in patients with type 2 diabetes (T2D) is related to the early onset of diabetic retinopathy [8]. Another study reported higher levels of TLR2 and TLR4 mRNAs in mononuclear cells of patients with type 1 diabetes compared to the control group [14]. The JAK/signal transducer and activator of transcription (STAT) pathway is an important signal transduction pathway in cells and can be used as downstream intermediates of various cytokines, hormones, and growth factors. With cytokine or growth factor signals, different combinations of JAK and STAT are activated. JAK/STAT protein regulates and maintains basic biological processes, including apoptosis, proliferation, immune response, and inflammation [15]. In patients with T2D, IL6-JAK2-STAT3 signal transfer in skeletal muscles plays a pathogenic role in inflammatory effects and the emergence of insulin resistance [16,17]. In vitro, STAT3 causes IL6 to induce TLR-4 production, leading to the expression of proinflammatory cytokines and insulin resistance in human skeletal muscle (myoblasts) [17]. Evidently, IL6 resistance appears in patients with T2D because IL6 can induce glycogen synthesis and glucose uptake in skeletal muscle, but not in the case of T2D [18]. Interestingly, compared with the normal control group, the muscle-specific STAT3 gene knockout mice did not show any changes in insulin sensitivity after being fed a high-fat diet [19]. Insulin resistance is the primary metabolic abnormality that leads to the development of prediabetes and T2D [20,21]. Insulin is involved in several homeostatic responses, including gluconeogenesis, regulation of renal blood flow, electrolyte/mineral balance, and vascular resistance [22]. Insulin and insulin receptor-mediated signaling are critical regulators of renal epithelial defenses [21]. We previously showed that glucose can enhance UPEC-induced infection in uroepithelial cells via TLR-4 and JAK/STAT1 signaling pathways [23]. Whether insulin can also reduce the severity of UPEC infection in bladder cells caused by high glucose levels is still unknown.

Here, we analyzed the expression of inflammatory factors under the influence of UPEC infection in bladder cells after insulin pretreatment in a high-glucose environment and explored the regulation of JAK/STAT transduction pathways in bladder cells during UPEC infection. The mechanism by which insulin reduces UPEC infection in bladder cells and the relationship between insulin receptor signaling, TLR-4-mediated transduction, and JAK/STAT-dependent pathways were also verified. The effects of different concentrations of insulin pretreatment in a high glucose environment on the expression of inflammatory factors and the regulation pathways during UPEC infection were elucidated, which may be helpful for the follow-up treatment of UTI in patients with diabetes.

## 2. Materials and Methods

### 2.1. Bacterial Strain

UPEC CFT073 from ATCC 700928 with fluorescent pGFP was used as the model organism [24]. Bacterial growth was determined spectrophotometrically at an optical density of 600 nm (OD_600_). For in vitro infection, the bacteria were suspended in culture medium at multiplicity of infection (MOI, cells: bacteria = 1:100).

### 2.2. Bladder Cell Culture, Insulin Pre-Treatment and UPEC Infection

Human normal bladder cell line SV-HUC-1 was cultured in F-12K Medium (Kaighn’s Modification of Ham’s F-12 Medium) containing 2 mM L-glutamine and 1500 mg/L sodium bicarbonate. (GIBCO-BRL #21127-022). Cultivate with 50 ng/mL bovine pituitary extract medium and place it in a 37 °C, 5% CO_2_ incubator. Replace the culture medium in 2–3 days. For infection experiment, 5.5 × 10^5^ SV-HUC-1 cells were incubated in cell culture medium with 15 mM glucose for 24 h, after replacing the medium, and then treated with insulin at different concentrations (5, 10, 20 and 40 ug/mL) for 24 h culture, then replaced the medium and infected with pGFP-UPEC (MOI cells: bacteria =1:100) for 4 h. After the 4 h incubation period, the infected monolayers were washed 4 times with phosphate-buffered saline (PBS) and incubated for 30 min in cell culture medium containing gentamicin (100 μg/mL; Sigma-Aldrich, St. Louis, MO, USA). Finally, the bacteria’s invasion ability was detected by immunofluorescence microscope and plate counting method. The untreated cells were used as the normal control group, and the SV-HuC-1 cells infected with pGFP-UPEC alone were used as the positive control group. The other two group of cells treated with the same treatment were added to Trizol or RIPA buffer for cryo-preservation after collection, respectively. The cell samples were collected for subsequent RNA nucleic acid and protein expression determination.

### 2.3. UPEC Infections and Quantifying

To measure bacterial invasion, the bladder cells were lysed and harvested using 0.5% tryp-sin (Gibco)–0.1% Triton X-100 (Amresco, Solon, OH, USA), and then plated the lyzed cell samples onto the NB medium containing Ampicillin. After 24 h incubation, the total colony-forming units (CFU) were counted to quantify the invaded bacteria within cells. To detect the green fluorescence of UPEC, SV-HUC-1 cells seeded onto 18 mm coverslip were infected with pGFP-UPEC (MOI of 50) and observed by fluorescence microscopy as described previously [25]. Briefly, SV-HUC-1 cells seeded onto coverslips in 6-well plates were treated as per previous description at 37 °C and 5% CO_2_. After treating, cells were washed four times with PBS and then cells were fixed and permeabilized with the BD Cytofix/Cytoperm solution for 30 min, the BD Perm/Wash buffer was used to wash the cells and to dilute the antibodies for following staining. Treated cells were washed and blocked with blocking solution (goat anti-human CD16/CD32 antibody/BSA, 1:19 in 1× PBS) for 30 min. The fluorescent markers used included Human anti-STAT1, STAT3 and insulin receptor (IR) phycoerythrin (PE)-conjugated antibody (1:500) (R&D Systems, Minneapolis, MN, USA) for 1 h incubation. After washing, the coverslips with stained cells were picked up and placed upside down on the slides, then visualized with a Leica TCS SP2 laser scanning confocal microscope (Leica, Bensheim, Germany) under 100 × magnifications. SV-HUC-1 cell nuclei were stained by incubation with 4′,6-diamidino-2-phenylindole (DAPI) diluted 1:1000 for 1 min. At least three coverslips per condition were examined. To quantify invasion, images of 20 random fields of each coverslip were acquired and counted with the help of computer image analysis software (ImageJ, Version 1.53n, NIH, Bethesda, MD, USA).

### 2.4. Cell Protein Extraction

The treated cells were centrifuged at 4 °C, 1800 rpm for 5 min, and the supernatant was poured out and placed in Protein lysis buffer (Pierce, Rockford, IL, USA). The cells were homogenized with a micro grinder and then mixed evenly to collect the expressed protein. After all the samples are collected, take them out and thaw them on ice, then centrifuge at 12,000 rpm at 4 °C for 10 min, and collect the supernatant to obtain total protein. Store in the refrigerator at −80 °C (after overnight) for later use.

### 2.5. Western Blotting

The proteins in the sample were harvested using RIPA lysis buffer (Merck Millipore, Burlington, MA, USA). The experimental procedure was also following our previous publication [24]. The antibodies used are listed as follows: anti-JAK1 (BD biosciences, East Rutherford, NJ, USA); anti-JAK2, anti-STAT1 (Cell Signaling, Beverly, MA, USA); anti-pSTAT1, anti-STAT3 (Cell Signaling, Beverly, MA, USA); anti-pSTAT3, anti-SOCS3 (Abcam, Cambridge, UK); anti-insulin receptor (Abcam, Cambridge, UK); anti-TLR-4 (Proteintech, Chicago, IL, USA); anti-IL-6 (Bioworld Technology, St Louis Park, MN, USA); and anti-β-actin (Santa Cruz, Santa Cruz, CA, USA) at room temperature (RT) for 1 h. After the incubation with the appropriate secondary horseradish peroxidase-conjugated IgG antibody (R&D Systems) for 30 min at RT, the protein bands on the membrane were detected with ECL-Plus Western Blot Detection system (GE Healthcare UK LTD) according to the instructions of the manufacturer. All experiments were replicated at least thrice.

### 2.6. RNA Isolation and RT-PCR

Total RNA was isolated from SV-HUC-1 cells using TRIzol^™^ reagent (Life Technologies, Carlsbad, CA, USA), then reverse transcribed to cDNA using a High-Capacity cDNA Reverse Transcription Kit (Applied Biosystems, Foster City, CA, USA). Real-time PCR (qPCR), performed with 2X SensiFAST^™^ SYBR^®^ No-ROX Kit (Bioline, London, UK), was analyzed using the Roche LightCycler^®^ 480 instrument (Roche Applied Science, Indianapolis, IN, USA). The experimental procedure and primer sequence were both following our previous publication [24]. The RT-PCR analysis was carried out in triplicate for each of the three independent samples.

### 2.7. Flow Cytometry

The cells were harvested using 0.25% trypsin-EDTA (Gibco/Invitrogen, Grand Island, NY, USA), which was neutralized with 10% soybean trypsin inhibitor (Gibco/Invitrogen, Grand Island, NY, USA). These cells were centrifuged at 1000× *g* for 10 min at 4 °C. The samples were washed and resuspended in 0.22 μm filtered PBS. After incubation with the human anti-STAT1, STAT3 and insulin receptor (IR) phycoerythrin (PE)-conjugated antibody (R&D Systems) for 30 min at RT, the samples were washed and resuspended in 0.22 μm filtered PBS. The intracellular pGFP-UPEC was monitored at 525 nm and PE labeling monoclonal antibodies were monitored at 575 nm using BD-FACS VERSE (BD Biosciences, CA, USA).

### 2.8. Statistical Analysis

The data were expressed as mean ± standard deviation (SD). Analysis of variance was used to analyze the differences between various treatment groups and controls. Statistical differences between groups were determined using the student’s *t* test. A *p* value of <0.05 was considered significant.

## 3. Results

### 3.1. Insulin Dose-Dependently Reduces UPEC Infection in Bladder Cells in a High-Glucose Environment

The effects of insulin on UPEC infection of bladder cells in a high-glucose environment were studied using a colony assay (Figure 1). After we diluted the cell lysed samples after bacterial infection by 100 and 10,000 times and performed the colony assay, the reduced results of insulin pretreatment were still the same or even more significant (Figure 1A1,B1, with 100× dilution; Figure 1A2,B2, with 10,000× dilution). Treatment of SV-HUC-1 cells with 15 mM glucose promoted UPEC infection relative to general infections (*p* < 0.01). Pretreatment with 20 and 40 μg/mL insulin for 24 h significantly and dose-dependently reduced UPEC infection in SV-HUC-1 cells compared to 15 mM glucose-treated cells (*p* < 0.01, as compared to 15 mM glucose group). Thus, insulin can dose-dependently reduce the ability of UPEC to infect bladder cells in a high-glucose environment.

### 3.2. Insulin Reduces UPEC Infection and Downregulates the Expression of STAT-1/3 and IR in SV-HUC-1 Cells

To clarify how insulin reduces UPEC infection in bladder cells in a high-glucose environment, SV-HUC-1 cells treated with different concentrations of insulin were co-cultured with GFP-UPEC, and infection and cell marker expression were assessed (STAT1, STAT3, and IR) using fluorescence microscopy and flow cytometry. SV-HUC-1 cells without treatment or UPEC infection as the negative control (Appendix A). Pre-treatment with 20 and 40 μg/mL insulin significantly reduced UPEC infection relative to the 15 mM glucose-treated group (*p* < 0.05) (Figure 2A,B and Figure 3A,B). Pretreatment with insulin also downregulated both STAT1 and STAT3 expression in bladder cells in a dose-dependent manner (Figure 2A,B and Figure 3A,B) (*p* < 0.05 and 0.001, respectively, as compared to the 15 mM glucose-treated group). However, the effect of insulin pretreatment on the expression of IR in SV-HUC-1 cells was not completely consistent with that on STAT1 and STAT3. Although compared with the 15 mM glucose treatment group, the 10 and 20 µg/mL insulin-treated groups showed downregulation of IR expression (*p* < 0.05), the effect was not dose-dependent, and the IR expression gradually increased with higher doses (Figure 4A,B).

The data obtained from the flow cytometric test showed similar results of both STAT1 and STAT3 expression in fluorescence microscopy, in which pretreatment with insulin dose-dependently decreased UPEC infection and expression of STAT1 and STAT3 in the bladder cells, especially in the 40 μg/mL-treated group (*p* < 0.05 and 0.01, respectively, as compared to the 15 mM glucose-treated group) (Figure 5A,B). In the flow cytometry experiment, insulin was also found to induce IR expression more obviously, but not significantly different from the glucose treatment group (Figure 5A,B). These results confirm that insulin not only reduces UPEC infection but also downregulates the expression of STAT1 and STAT3 in bladder cells, unlike IR expression.

### 3.3. Insulin-Mediated Reduction of UPEC Infection in Bladder Cells Is Related to Downregulation of the JAK/STAT1 Signaling Pathway and TLR-4 Expression

To verify whether insulin reduced UPEC infection in bladder cells through TLR-4 mediation and JAK/STAT1 signaling pathway in a high-glucose environment, the expression of related mRNAs and proteins was evaluated by qPCR and WB. Figure 6 shows the expression of all the genes in UPEC-infected SV-HUC-1 cells with/without insulin pretreatment, including IR (IRα/β), TLR-4, inflammatory IL-6 and IFN-γ cytokines, and JAK1/2 and STAT1/3 in JAK/STAT signaling pathways. In our previous study, the expression of TLR-4 in the 15 mM glucose-treated group was similar to that of the general infected cells, whereas a significant difference with/without glucose treatment was observed in the other genes such as JAK1/2, STAT1/3, and inflammatory cytokines IL-6 and IFN-γ (*p* < 0.01). Furthermore, except for IRα/β, the expression of all genes showed a significant and dose-dependent downregulation after 10–40 μg/mL insulin pretreatment (*p* < 0.001, as compared to the 15 mM glucose-treated groups). The protein expression experiments further showed that insulin decreased UPEC infection in bladder cells in a high-glucose environment not only by downregulating the JAK/STAT signaling pathway, including the expression of JAK1/2, STAT-1/3, and phosphorylated STAT-1/3, but also by decreasing the expression of TLR-4 proteins and pro-inflammatory IL-6 (Figure 7A,B), especially in the 20 and 40 μg/mL-pretreated groups (*p* < 0.05, compared to the 15 mM glucose-treated groups). SOCS3, a STAT inhibitor, was slightly and dose-dependently induced by insulin pretreatment in UPEC-infected cells, and the effect was reverse-relative to the results of phosphorylated STAT-1/3 expression. Moreover, insulin did not seem to increase or decrease IR expression in comparison to the glucose-treated group (Figure 7B). In summary, the levels of protein expression were consistent with those of mRNA; TLR-4 protein, JAK/STAT-related transduction factors, or inflammation-related cytokines, all appeared to be downregulated by insulin. Thus, insulin plays an important role in reducing inflammatory responses in UPEC-infected bladder cells by inhibiting the JAK/STAT signaling pathway, especially by downregulating STAT1, STAT3, and TLR-4.

## 4. Discussion

Previously, we verified that glucose promotes UPEC-induced cystitis and invasion into uroepithelial epithelial cells by activating TLR-4-mediated UPEC infection and JAK/STAT1-dependent pathways [23]. However, whether insulin, which plays an important role in glucose homeostasis and the treatment of T2D, can effectively improve UPEC infection of urothelial epithelial cells in a high-glucose environment remains unclear. Here, we showed that insulin could effectively reduce UPEC infection in bladder cells by downregulating the JAK/STAT signaling pathway in a high-glucose environment.

Insulin and IR-mediated signaling are critical regulators of uroepithelial defenses. IR deletion and insulin resistance increase UTI risk by suppressing the phosphatidylinositide 3-kinase signaling pathway (PI3K/AKT) and downstream antimicrobial peptide expression [21]. Insulin induces the activation of PI3K/AKT to shield urothelial cells from UPEC [26]. Our previous results also indicated that STAT1 and STAT3 play critical roles in the regulation of UPEC invasion and infection in uroepithelial cells, especially those pretreated with glucose [23,27]. Therefore, it is worthwhile to understand the synergistic effects of these insulin signaling pathways in UPEC-infected bladder cells. Here, we showed that insulin can effectively interfere with UPEC infection within bladder cells. SV-HUC-1 cells pretreated with 20 and 40 μg/mL insulin showed significantly reduced UPEC infection rates and colonized numbers post-infection compared to the 15 mM glucose-treated group (Figure 1). Insulin pretreatment not only reduced UPEC infection within the cells but also suppressed the STAT1 and STAT3 protein expression in a dose-dependent manner. Both STAT1 and STAT3 proteins are involved in the process of glucose-enhanced UPEC infection of bladder cells. Their expression was downregulated in a dose-dependent manner and synchronized with the infection rate (Figure 2, Figure 3 and Figure 4). However, the IR expression in the UPEC-infected bladder cells pretreated with insulin and 15 mM glucose did not reveal similar results to those of STAT1 or STAT3; it was significantly downregulated initially in the 10 μg/mL insulin-pretreated group and gradually increased with higher doses of insulin. Insulin itself is arguably the most important negative regulator of insulin signaling. IR undergoes ligand-induced internalization, lysosomal degradation, or recycling to the cell surface. As insulin levels rise, receptor downregulation at the cell surface paradoxically decreases insulin signaling, which is promptly reversed when insulin levels fall due to glucose reduction [28]. However, whether IR expression is the same in a high-glycemic environment or in an in vitro cell environment needs to be further clarified.

The integrative role of insulin in regulating the immune system not only includes its direct effects on the immune system but also the fact that inflammatory mediators often activate signaling pathways that crosstalk with canonical insulin pathways. Both human and mouse pancreatic beta-cells express TLR4, and a classic TLR4 agonist, LPS, accentuates glucose-stimulated insulin release [29,30]. The JAK/STAT pathways, which are downstream effectors of various cytokines, also exemplify the effect of insulin on the immune system. For example, JAK2 activation potentiates PI3K activity via insulin receptor substrate (IRS)-2 activation following leptin exposure [31]. Similarly, JAK1, -2, and -3 have been shown to activate IRS-1, as IL-13R activates both the JAK/STAT and PI3K/AKT pathways [32]. Our data demonstrated that insulin may reduce UPEC infection and inhibits the JAK/STAT signaling pathway in bladder cells by reducing the expression of TLR4 and pro-inflammatory cytokines. Although additional experiments such as preventing these signaling from transducing and test whether UPEC infection is then reduced are needed to perform for following verification. Both mRNA and protein levels showed dose-dependent reductions in a high-glucose environment with insulin pretreatment (Figure 5 and Figure 6). Long-term UPEC infection can suppress cytokine signaling (SOCS) protein to inhibit pro-inflammatory signaling mediated by JAK/STAT pathway inactivation in human uroepithelial cells [33,34]. Our results also showed that SOCS3 protein expression was inversely correlated with the JAK and STAT proteins. STAT inactivation with an increase in SOCS3 may act as a protective mechanism to limit pro-inflammatory cytokines such as IL-6 and IFN-γ in the urothelium to avoid excessive UTI [35,36]. In addition, IR expression (IR-α or IR-β) did not show any significant difference after insulin pretreatment compared to the 15 mM glucose-treated group, even in the 40 μg/mL group.

Insulin stimulation of glucose transporter (GLUT)-mediated glucose transport in cells is essential for the maintenance of glucose homeostasis. Marked reduction of GLUT4, the predominant glucose transporter, causes insulin resistance and increases the risk of diabetes. The stimulatory effect of insulin in insulin-sensitive tissues results from GLUT4 translocation through the membrane and markedly augments glucose transport into the cell [37]. Another study reported that the expression levels of virulence genes in UPEC were significantly reduced in the presence of insulin and/or glucose [38]. Therefore, insulin may reduce the adhesion and invasion of UPEC to bladder cells by affecting glucose homeostasis inside and outside the cell, and decrease the augmentation of JAK/STAT pathway-mediated inflammation and infection in bladder cells in a high-glucose environment. However, the crosstalk between insulin signaling and JAK/STAT pathway requires further clarification.

## 5. Conclusions

The burden of senescent cells in preadipocytes has been linked to inflammation, insulin resistance, and T2D. STAT1 and STAT3 phosphorylation in these cells determine their antagonistic functions in regulating growth arrest and inflammation [39]. In this study, we provide insights into how STAT1/STAT3 signaling coordinates UPEC infection-induced inflammation through functional interactions with the insulin-signaling pathway. Our study demonstrated that insulin not only reduced UPEC infection in bladder epithelial cells but also decreased the JAK/STAT transduction pathway during infection in a high-glucose environment. However, the mechanism of action of insulin as an anti-UPEC infection mediator in bladder cells with high glucose pretreatment through the JAK/STAT signaling pathway requires further research. Taken together, this study provides a possible direction for insulin to be beneficial in treating clinical UPEC infection in patients with T2D.

## Figures and Tables

**Figure 1 microorganisms-09-02421-f001:**
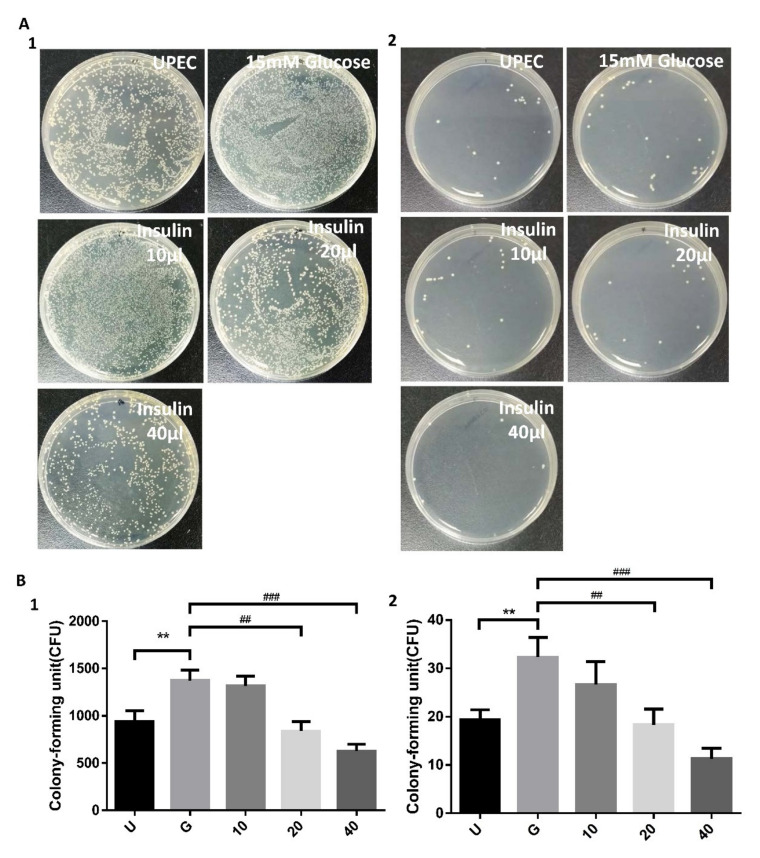
Influence of insulin on UPEC infection in bladder cells in a high glucose environment. SV-HUC-1 cells were pretreated with different concentrations of insulin and 15 mM glucose for 24 h and then infected with pGFP-UPEC (MOI:1:100) as described in the text. UPEC infection in bladder cells was examined by plating on NB medium agar (**A**) and determined using Image J software (**B**). Colony forming units (CFUs) were acquired after plating out lysed solutions of infected cells. The counting results of the lysate diluted 100 and 10,000 times were shown in (**A1**), (**B1**) and (**A2**), (**B2**), respectively. Data are expressed as the mean ± SD of three independent experiments. The definitions of each letter/number represented by the X axis are: U: UPEC infection alone; G: with 15 mM glucose pretreatment; 10, 20 and 40: pretreated with 10, 20 and 40 μg/mL insulin, respectively + 15 mM glucose. ** *p* < 0.01, compared with the infection control group. ^##^ *p* < 0.01, ^###^ *p* < 0.001, compared with the 15 mM glucose pretreated group.

**Figure 2 microorganisms-09-02421-f002:**
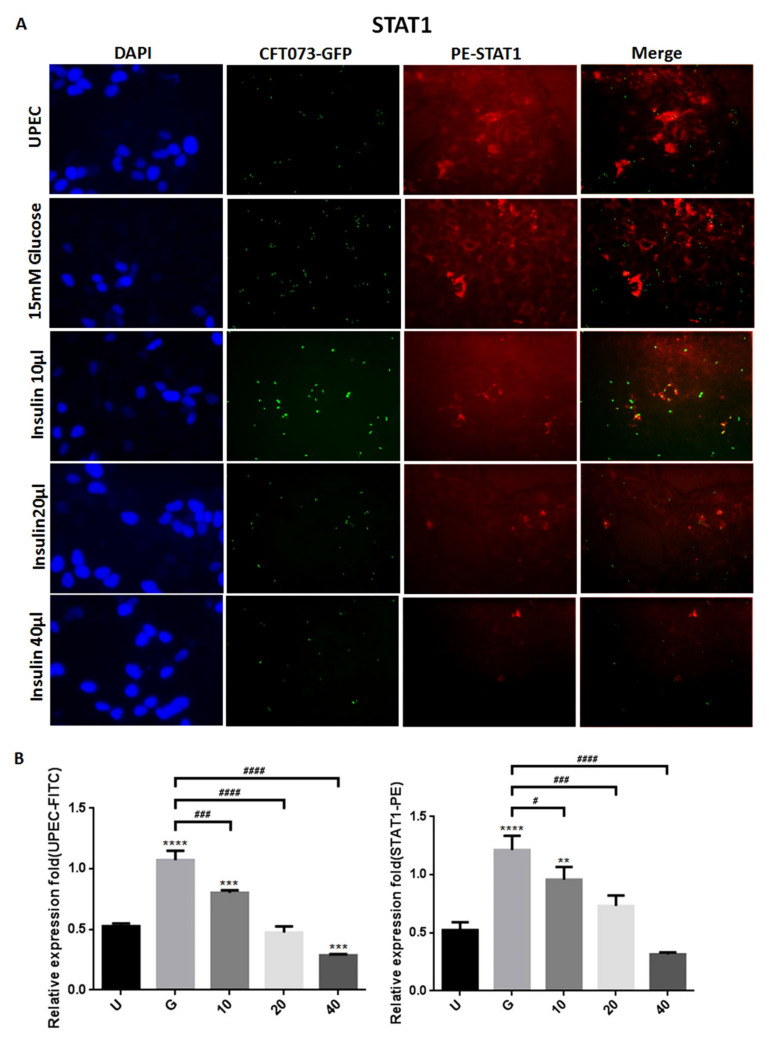
Insulin affected UPEC infection and STAT1 expression in bladder cells. After 24 h of pretreatment with different concentrations of insulin and 15 mM glucose, GFP-UPEC and PE-STAT1 expression in SV-HUC-1 cells post-infection were (**A**) observed by fluorescence microscopy and (**B**) measured using ImageJ software. Cells infected with UPEC alone were used as the positive controls. The data shown are representative of a typical result. DAPI was used to count the number of cells and as a standard for calculating the fluorescence expression ratio of cells. Cell images were captured using a microscope (Leica) at 100× magnification. Data are expressed as mean ± SD from three separate experiments. The definitions of each letter/number represented by the X axis are: U: UPEC infection alone; G: with 15 mM glucose pretreatment; 10, 20 and 40: pretreated with 10, 20 and 40 μg/mL insulin, respectively + 15 mM glucose. ** *p* < 0.01, *** *p* < 0.001, **** *p* < 0.0001, compared to positive control groups. ^#^ *p* < 0.05, ^###^ *p* < 0.001, ^####^ *p* < 0.0001, compared between the two groups.

**Figure 3 microorganisms-09-02421-f003:**
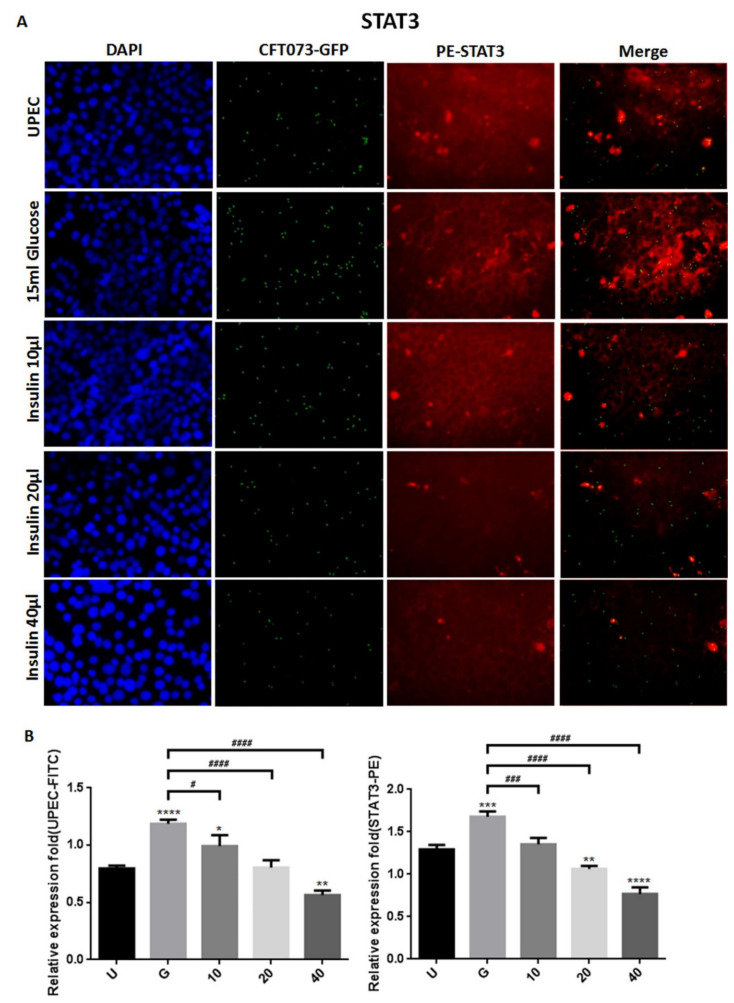
Insulin affected UPEC infection and STAT3 expression in bladder cells. After 24 h of pretreatment with different concentrations of insulin and 15 mM glucose, GFP-UPEC and PE-STAT3 expression in SV-HUC-1 cells post-infection were (**A**) observed by fluorescence microscopy and (**B**) measured using ImageJ software. Cells infected with UPEC alone were used as the positive controls. The data shown are representative of a typical result. DAPI was used to count the number of cells and as a standard for calculating the fluorescence expression ratio of cells. Cell images were captured using a microscope (Leica) at 100× magnification. Data are expressed as mean ± SD from three separate experiments. The definitions of each letter/number represented by the X axis are: U: UPEC infection alone; G: with 15 mM glucose pretreatment; 10, 20 and 40: pretreated with 10, 20 and 40 μg/mL insulin, respectively + 15 mM glucose. * *p* < 0.05; ** *p* < 0.01, *** *p* < 0.001, **** *p* < 0.0001, compared to positive control groups. ^#^ *p* < 0.05, ^###^ *p* < 0.001, ^####^ *p* < 0.0001, compared between the two groups.

**Figure 4 microorganisms-09-02421-f004:**
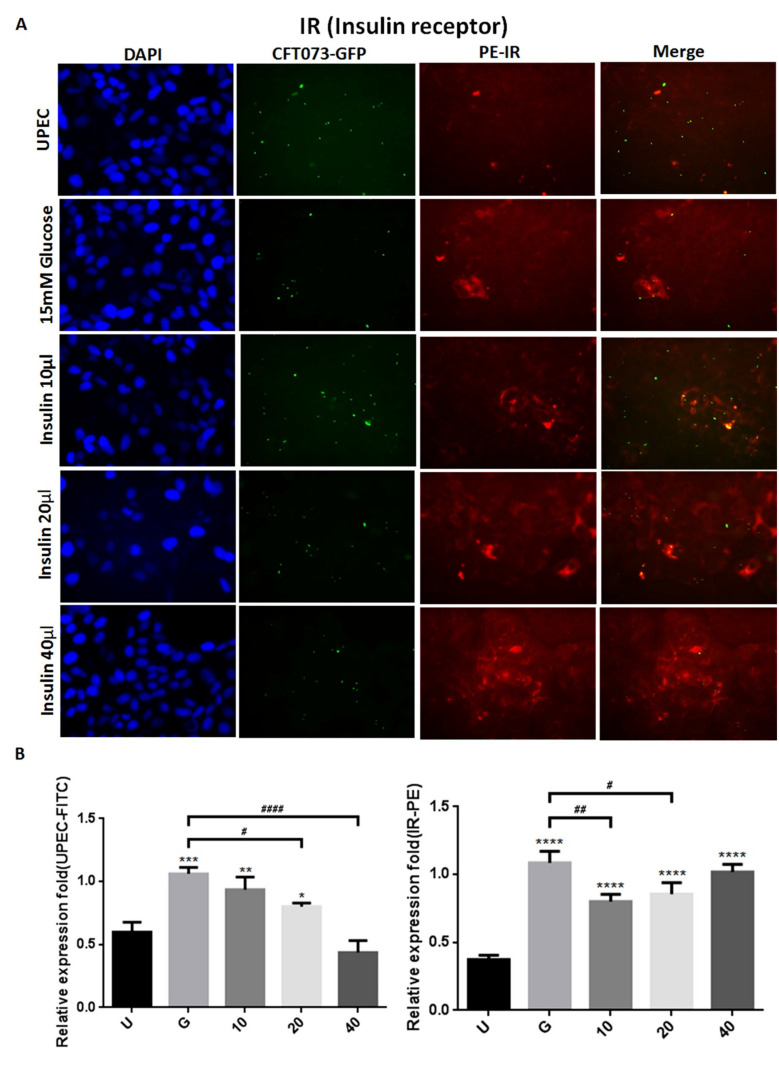
Insulin affected UPEC infection and insulin receptor expression in bladder cells. After 24 h of pretreatment with different concentrations of insulin and 15 mM glucose, GFP-UPEC and PE- insulin receptor (IR) expression in SV-HUC-1 cells post-infection were (**A**) observed by fluorescence microscopy and (**B**) measured using ImageJ software. Cells infected with UPEC alone were used as the positive controls. The data shown are representative of a typical result. DAPI was used to count the number of cells and as a standard for calculating the fluorescence expression ratio of cells. Cell images were captured using a microscope (Leica) at 100× magnification. Data are expressed as mean ± SD from three separate experiments. The definitions of each letter/number represented by the X axis are: U: UPEC infection alone; G: with 15 mM glucose pretreatment; 10, 20 and 40: pretreated with 10, 20 and 40 μg/mL insulin, respectively + 15 mM glucose. * *p* < 0.05; ** *p* < 0.01, *** *p* < 0.001, **** *p* < 0.0001, compared to positive control groups. ^#^ *p* < 0.05, ^##^ *p* < 0.01, ^####^ *p* < 0.0001, compared between the two groups.

**Figure 5 microorganisms-09-02421-f005:**
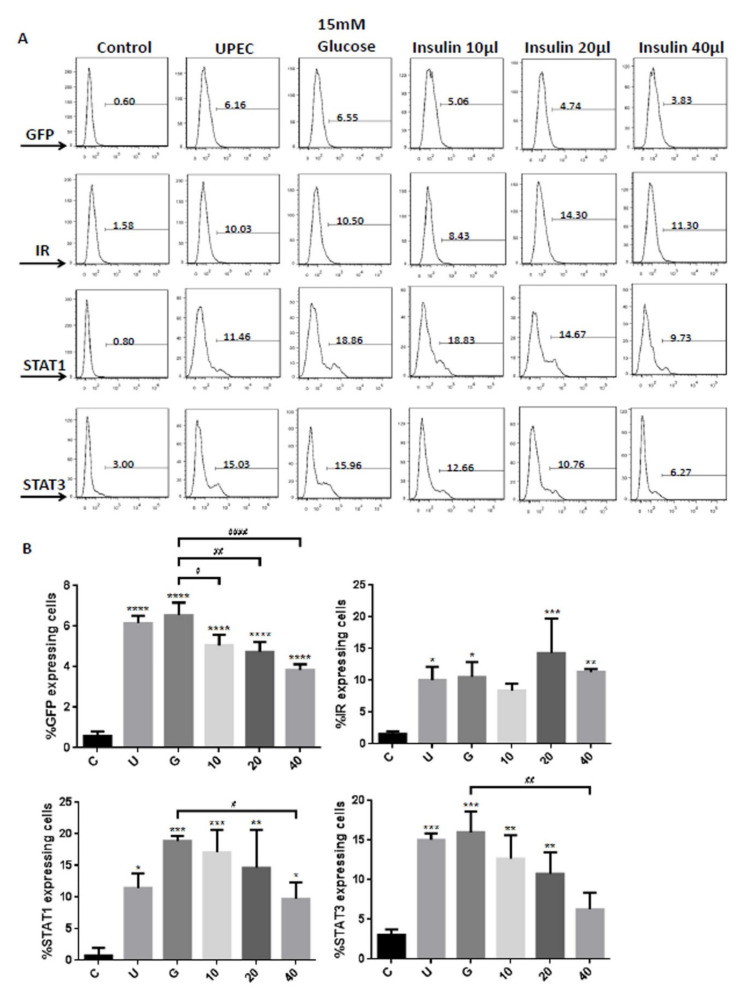
Insulin downregulated insulin receptor, STAT1, and STAT3 expression in UPEC-infected bladder cells. Flow cytometry data were obtained from insulin-pretreated UPEC-infected SV-HUC-1 cells in a high-glucose environment. (**A**) The frequency of GFP, insulin receptor (IR), STAT1, and STAT3 expression in uninfected and UPEC-infected bladder cells with or without different concentrations of insulin (10, 20, and 40 µg/mL), and 15 mM glucose pretreatment for 24 h is shown as a histogram, and the results shown are representative of typical results. Numbers show the average frequency of GFP-, IR-, STAT1-, or STAT3 expressing cells. (**B**) Bars depict the data from three separate experiments and are expressed as mean ± SD. Cells with UPEC-infected cells alone were used as a positive control. The definitions of each letter/number represented by the X axis are: C: control; U: UPEC infection alone; G: with 15 mM glucose pretreatment; 10, 20 and 40: pretreated with 10, 20 and 40 μg/mL insulin, respectively + 15 mM glucose. * *p* < 0.05; ** *p* < 0.01; *** *p* < 0.001, **** *p* < 0.0001, compared to positive control groups. ^#^ *p* < 0.05, ^##^ *p* < 0.01, ^####^ *p* < 0.0001, compared between the two groups.

**Figure 6 microorganisms-09-02421-f006:**
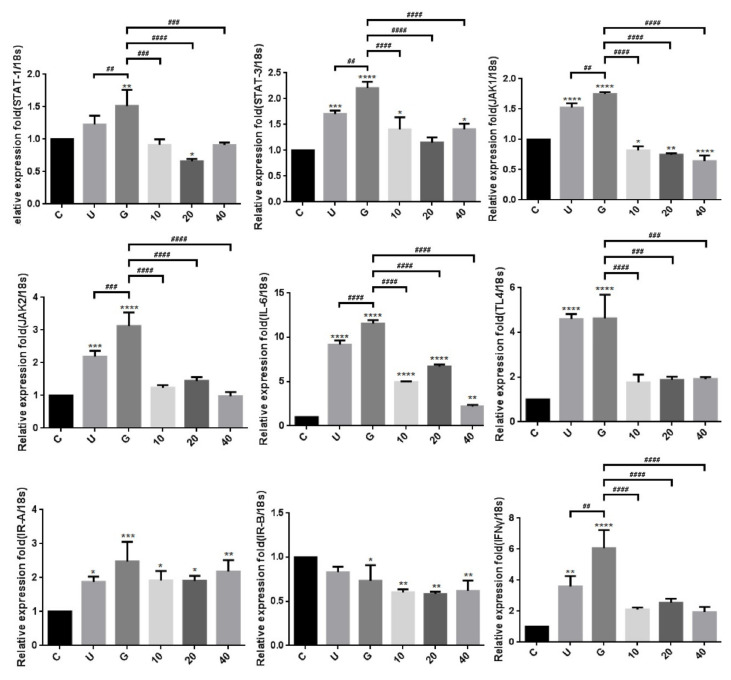
Transcriptional effects of insulin on JAK/STAT1 signaling pathway in UPEC-infected bladder cells in a high-glucose environment. Total RNA was isolated from treated cells and analyzed using qPCR. The mRNA levels of JAK1, JAK2, STAT1, STAT3, IFN-γ, IL-6, TLR-4, and IR-α/β were detected in UPEC-infected bladder cells with different concentrations of insulin and 15 mM glucose pretreatment for 24 h. The definitions of each letter/number represented by the X axis are: C: control; U: UPEC infection alone; G: with 15 mM glucose pretreatment; 10, 20 and 40: pretreated with 10, 20 and 40 μg/mL insulin, respectively + 15 mM glucose. Data from three separate experiments are expressed as mean ± SD. * *p* < 0.05; ** *p* < 0.01; *** *p* < 0.001, **** *p* < 0.0001, compared to positive control groups. ^##^ *p* < 0.01, ^###^ *p* < 0.001, ^####^ *p* < 0.0001, compared between the two groups.

**Figure 7 microorganisms-09-02421-f007:**
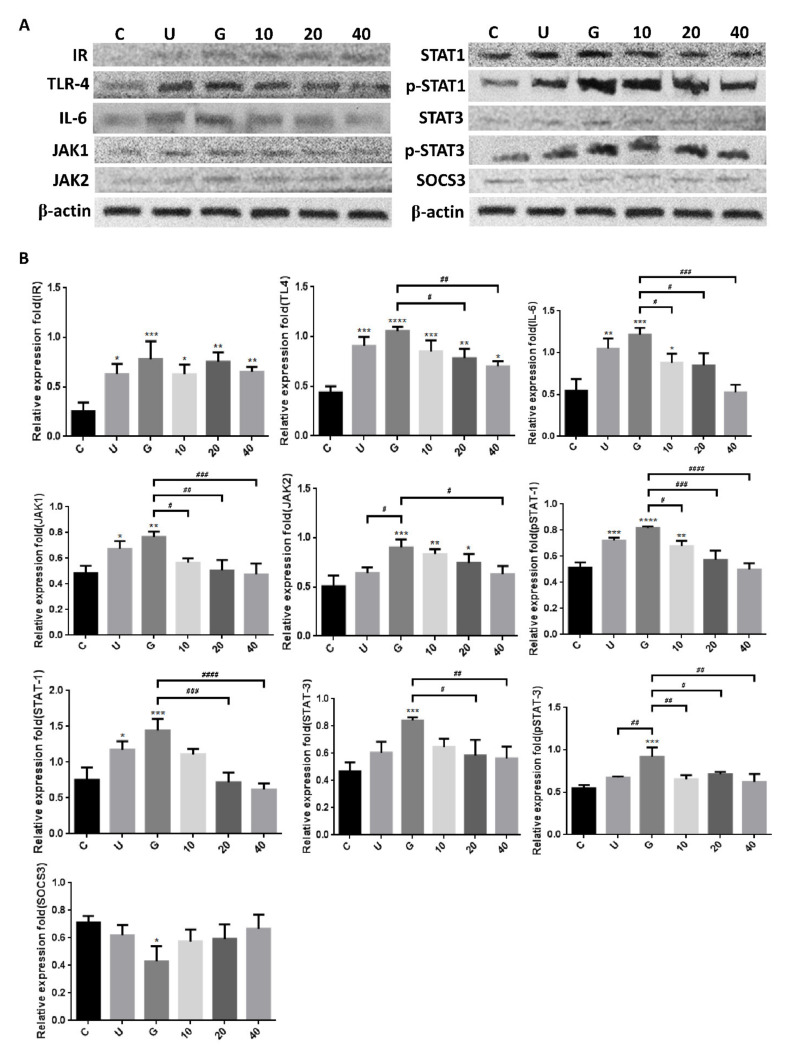
Insulin downregulated the expression of proteins associated with the JAK/STAT1 signaling pathway and inflammation in UPEC-infected bladder cells in high-glucose environment. SV-HUC-1 cells were pretreated with 10, 20, and 40 μg/mL insulin and 15 mM glucose for 24 h, and then infected with UPEC. Total protein from all the cell groups was collected for detection. (**A**) Total protein expression of JAK1, JAK2, STAT1, STAT3, phosphorylated-STAT1/STAT3 and the inhibitor, SOCS3, TLR-4, IL-6, and insulin receptor (IR) were analyzed by western blotting. (**B**) All data were normalized to the internal reference β-actin. Results were assessed using a densitometer and quantified using ImageJ software (NIH). The definitions of each letter/number represented by the X axis are: C: control; U: UPEC infection alone; G: with 15 mM glucose pretreatment; 10, 20 and 40: pretreated with 10, 20 and 40 μg/mL insulin, respectively + 15 mM glucose. The results are presented as the mean ± SD of three independent experiments. * *p* < 0.05; ** *p* < 0.01; *** *p* < 0.001, **** *p* < 0.0001, compared to positive control groups. ^#^ *p* < 0.05, ^##^ *p* < 0.01, ^###^ *p* < 0.001, ^####^ *p* < 0.0001, compared between the two groups.

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
