# Peer review of "Insulin Downregulated the Infection of Uropathogenic Escherichia coli (UPEC) in Bladder Cells in a High-Glucose Environment through JAK/STAT Signaling Pathway"

_microorganisms, 2021, doi:10.3390/microorganisms9122421_

Round 1
Reviewer 1 Report
The authors seek to dissect the complex host-pathogen interactions that render diabetes more susceptible to UTI compared to non-diabetic people. Specifically, the authors test the effect of insulin in a high-glucose environment on the immune response to UPEC infection using an in vitro cell culture system. The authors data reveal important insights into the downstream regulation of inflammatory pathways following infection, in an environment that mimics diabetes. While these insights are novel, enthusiasm for the manuscript was diminished by the abbreviated materials and methods section, which contained confusing language at times on how specifically each experiment was performed. Additionally, several key issues should be addressed to strengthen the conclusions of the paper:
Major:
Materials and Methods:
Line 98: what “culture medium” are the authors referring to? LB? Tissue culture medium?
Lines 98-99: The authors should clarify what MOI was used for the experiments.
Lines 100-113: As written it is unclear how the authors are performing the bladder cell culture infections. The authors should clarify how this experiment is performed by adding additional details. Specifically, are the authors incubating the bacteria in culture medium with excess glucose and then adding insulin? After incubation are the bacteria then added to the bladder cells at an MOI 1:100? Additionally, the authors state that “cells” infected with pGFP-UPEC only were used as positive controls. Are these bacteria grown with glucose? Is there a control where the bacteria are grown without glucose? Are there bacteria only grown in the presence of insulin, but without glucose?
Line 120: The authors state that CFUs were enumerated to quantify bound bacteria, but these cells/bacteria were treated with gentamicin, which should kill extracellular/bound bacteria. Are the authors actually measuring invaded bacteria?
Line 123-124: the authors should briefly describe how the bladder cells on coverslips were stained.
Results:
Lines 183-184: the figure legend states that the HUC cells were exposed to glucose and insulin, but the way the materials and methods is written it implies the bacteria are the cells that are exposed. The authors should clarify how this experiment is performed.
Section 3.1: It is unclear how the authors are generating accurate CFU counts from plates that essentially have bacterial lawns. The authors should perform serial dilutions, count plates that have between 30-300 colonies, and calculate the appropriate CFUs from the dilution factor.
Section 3.2: It would give clarity to the results if the authors could include uninfected bladder cells as a comparison. Additionally, including an uninfected insulin only control would also strengthen the conclusions of the paper.
Section 3.3: It would provide clarity for the reader if the authors could include definitions of what each letter/number I the X-axis represents in the figure legend for figures 6 and 7.
Minor:
Abstract:
Lines 18-19: are the authors specifically talking about hospitalized individuals with diabetes? Or are they referring to individuals with diabetes having higher rates of UTI than individuals without diabetes?
Line 26: what does “IR” refer to?
Line 42: What do the authors mean when they say “female genitals often introduce cystitis bacteria?”
Line 53 and 56: E. coli do not encode cilia. Cilia are only encoded by mammalian cells. Do the authors mean pili? Or are they talking about flagella? Or do the authors mean E. coli adhere to cilia on the surface of epithelial cells?
Line 73: “In vitro” should be italicized
Materials and Methods:
Lines 137 and 153: “experiment” should read “experimental”
Results:
Figure 5 is discussed before figure 4.
Discussion:
Line 345: “UTi” the “i” should be capitalized
Author Response
11 03, 2021
Dear Editor:
Attached please find our revision manuscript entitled “Insulin downregulated the infection of uropathogenic Escherichia coli (UPEC) in bladder cells in a high-glucose environment through JAK/STAT signaling pathway “and the Manuscript Number of our paper is microorganisms-1417429.
We greatly appreciate all the valuable comments that have offered. These comments have helped us in improving the quality of the manuscript. The revisions are listed below for review, the changes we have made are highlighted with BOLD word in the text and their locations in the manuscript are indicated in response of each problem. Please review the updated manuscript and kindly let us know if additional changes are necessary. We sincerely hope that the revised manuscript and this letter will furnish sufficient explanation for the reviewer’s questions.
Sincerely yours,
Po-Ching Cheng, Ph. D.
Department of Molecular Parasitology and Tropical Diseases,
School of Medicine, College of Medicine,
Taipei Medical University,
Taipei, Taiwan
Reviewer1:
Comments and Suggestions for Authors
The authors seek to dissect the complex host-pathogen interactions that render diabetes more susceptible to UTI compared to non-diabetic people. Specifically, the authors test the effect of insulin in a high-glucose environment on the immune response to UPEC infection using an in vitro cell culture system. The authors data reveal important insights into the downstream regulation of inflammatory pathways following infection, in an environment that mimics diabetes. While these insights are novel, enthusiasm for the manuscript was diminished by the abbreviated materials and methods section, which contained confusing language at times on how specifically each experiment was performed. Additionally, several key issues should be addressed to strengthen the conclusions of the paper:
We thank the reviewer for finding our manuscript novel and important.
Major:
Materials and Methods:
Line 98: what “culture medium” are the authors referring to? LB? Tissue culture medium?
Ans: Here we used SV-HUC-1 cell culture medium (F-12K Medium) without antibiotics as the medium for in vitro UPEC-SV-HUC-1 cell infection test.
Lines 98-99: The authors should clarify what MOI was used for the experiments.
Ans: Thank you for the comment. We used MOI 1:100 for our infection experiments as our previous study. The relative information was described in line 107 of section 2.2. We also added the brief description in line 99 of section 2.1.
Lines 100-113: As written it is unclear how the authors are performing the bladder cell culture infections. The authors should clarify how this experiment is performed by adding additional details. Specifically, are the authors incubating the bacteria in culture medium with excess glucose and then adding insulin? After incubation are the bacteria then added to the bladder cells at an MOI 1:100? Additionally, the authors state that “cells” infected with pGFP-UPEC only were used as positive controls. Are these bacteria grown with glucose? Is there a control where the bacteria are grown without glucose? Are there bacteria only grown in the presence of insulin, but without glucose?
Ans: We thank the reviewer for pointing this out. Sorry for the misunderstanding caused by the unclear description, we have revised the line 100-113 part of section 2.2 to avoid confusion. Actually, we incubated the bladder cells in culture medium with 15mM glucose for 24 h and then cultured with insulin for 24 h, then just added bacteria at an MOI 1:100. Additionally, we used SV-HuC-1 cells infected with pGFP-UPEC alone as positive controls. We pretreated with glucose or insulin are for the SV-HuC-1 bladder cells, not for the bacteria.
Line 120: The authors state that CFUs were enumerated to quantify bound bacteria, but these cells/bacteria were treated with gentamicin, which should kill extracellular/bound bacteria. Are the authors actually measuring invaded bacteria?
Ans: Yes, we are sorry for the vague description has caused a misunderstanding again. We actually want to measure the UPEC bacteria that infect into the bladder cells, not the adhered extracellularly bacteria. We have revised the line 115-121 of 2.3 section to avoid confusion.
Line 123-124: the authors should briefly describe how the bladder cells on coverslips were stained.
Ans: Thank you for your comment. We have added a brief description of cell staining in lines 125-137 in section 2.3 for clear understanding of the experimental procedure.
Results:
Lines 183-184: the figure legend states that the HUC cells were exposed to glucose and insulin, but the way the materials and methods is written it implies the bacteria are the cells that are exposed. The authors should clarify how this experiment is performed.
Ans: We thank the reviewer for pointing this out. Sorry again for the misunderstanding caused by the unclear description, we have revised the materials and methods of section 2.2 to avoid confusion. Actually, we incubated the bladder cells in culture medium with 15mM glucose for 24 h and then cultured with insulin for 24 h, then just added bacteria at an MOI 1:100. We pretreated with glucose or insulin are for the SV-HuC-1 bladder cells, not for the bacteria. Hope these explanations could solve reviewer’s confusion.
Section 3.1: It is unclear how the authors are generating accurate CFU counts from plates that essentially have bacterial lawns. The authors should perform serial dilutions, count plates that have between 30-300 colonies, and calculate the appropriate CFUs from the dilution factor.
Ans: Thank you for your comment. We have added a new data that used 100X dilutions of cell lysis samples on plate counting assay in section 3.1 and Figure 1 (line 192-195, 202-204; Figure 1A2 and 1B2), to clearer measure of the accurate CFU counts from plates.
Section 3.2: It would give clarity to the results if the authors could include uninfected bladder cells as a comparison. Additionally, including an uninfected insulin only control would also strengthen the conclusions of the paper.
Ans: Thank you for your comment. We have provided a supplemental data that include uninfected bladder cells as the supplemental information (line 401-404; supplemental information S1), to clearer explain the situation of PE-STAT1, PE-STAT3 and PE-IR expressions in infected cells. Additionally, our previous study (reference 23) have revealed that in the absence of UPEC infection, cells pretreated with glucose alone will not affect TLR-4 expression and JAK/STAT1 pathways. Therefore, in this study, we hope to continue to focus on the effect of insulin on UPEC-infected cells in a high-glucose environment. However, we believe that insulin-related discussions can be carried out in our follow-up experiments.
Section 3.3: It would provide clarity for the reader if the authors could include definitions of what each letter/number I the X-axis represents in the figure legend for figures 6 and 7.
Ans: We thank the reviewer for pointing this out. We have added the definition of each letter/number represented by the X axis to the legends in Figure 6 and Figure 7 (line 299-301, 311-313), and hope to provide clearer information.
Minor:
Abstract:
Lines 18-19: are the authors specifically talking about hospitalized individuals with diabetes? Or are they referring to individuals with diabetes having higher rates of UTI than individuals without diabetes?
Ans: We are referring to individuals with diabetes having higher rates of UTI than individuals without diabetes here.
Line 26: what does “IR” refer to?
Ans: Thanks for pointing out the mistake. "IR" here refers to the insulin receptor, we have revised the full name as it first appeared in the text.
Line 42: What do the authors mean when they say “female genitals often introduce cystitis bacteria?”
Ans: Here we mean that female genitals often have cystitis bacteria remained and have a high risk of possible infection of the urinary tract.
Line 53 and 56: E. coli do not encode cilia. Cilia are only encoded by mammalian cells. Do the authors mean pili? Or are they talking about flagella? Or do the authors mean E. coli adhere to cilia on the surface of epithelial cells?
Ans: Thanks for pointing out the mistake. Here we mean pili, we have revised the error to pili.
Line 73: “In vitro” should be italicized
Ans: Thanks for pointing out the mistake. We have revised the error in the text.
Materials and Methods:
Lines 137 and 153: “experiment” should read “experimental”
Ans: Thanks for pointing out the mistake. We have revised the error in the text.
Results:
Figure 5 is discussed before figure 4.
Ans: Because we wanted to compare the results of STAT1 and STAT3 between fluorescence microscopy and flow cytometry tests in this paragraph, her we discuss part data of figure 5 first with figure 3 results. In next paragraph, we also discussed the figure 4 data with part results of figure 5 at the same time.
Discussion:
Line 345: “UTi” the “i” should be capitalized
Ans: Thanks for pointing out the mistake. We have revised the error in the text.

Reviewer 2 Report
The methods used are sufficiently documented and allow replication of the study. Results obtained are well presented and data interpretation is also correct. Conclusion is well above the average.
In the title author should strictly follow the microbiological rule of writing the organism name
Abbreviations should be given only for the first time in bracket, thereafter given the abbreviated form, eg. UTI
The introduction should begin by technically defining UTI rather than cystitis, followed by classification, epidemiology and predisposing factors.
In the introduction authors mentioned ‘Bacterial cystitis can cause infections in women after sexual intercourse’. The word women should be modify, becoz it affects only young sexually active females and name of the disease is Acute Urethral Syndrome.
In few cases it may be caused by gonococcus, Chlamydia, herpes simplex virus, etc
What the author think about thisɁ
And what about the Pregnancy and other risk factor ….
Most cases of cystitis are caused by uropathogenic Escherichia coli (UPEC). Provide proper citation to this
Some studies can be precisely written as ‘a couple of studies’
Materials and methods, results and discussions are well written.
Author Response
11 03, 2021
Dear Editor:
Attached please find our revision manuscript entitled “Insulin downregulated the infection of uropathogenic Escherichia coli (UPEC) in bladder cells in a high-glucose environment through JAK/STAT signaling pathway “and the Manuscript Number of our paper is microorganisms-1417429.
We greatly appreciate all the valuable comments that have offered. These comments have helped us in improving the quality of the manuscript. The revisions are listed below for review, the changes we have made are highlighted with BOLD word in the text and their locations in the manuscript are indicated in response of each problem. Please review the updated manuscript and kindly let us know if additional changes are necessary. We sincerely hope that the revised manuscript and this letter will furnish sufficient explanation for the reviewer’s questions.
Sincerely yours,
Po-Ching Cheng, Ph. D.
Department of Molecular Parasitology and Tropical Diseases,
School of Medicine, College of Medicine,
Taipei Medical University,
Taipei, Taiwan
Reviewer2:
Comments and Suggestions for Authors
The methods used are sufficiently documented and allow replication of the study. Results obtained are well presented and data interpretation is also correct. Conclusion is well above the average.
Ans: We thank the reviewers for complimenting our manuscript.
In the title author should strictly follow the microbiological rule of writing the organism name
Abbreviations should be given only for the first time in bracket, thereafter given the abbreviated form, eg. UTI
Ans: Thanks for pointing out the mistake. We have revised the mistakes in our text.
The introduction should begin by technically defining UTI rather than cystitis, followed by classification, epidemiology and predisposing factors.
Ans: Thank you for the comment. We have revised the related description in our text (line 37-48).
In the introduction authors mentioned ‘Bacterial cystitis can cause infections in women after sexual intercourse’. The word women should be modify, becoz it affects only young sexually active females and name of the disease is Acute Urethral Syndrome.
Ans: Thanks for pointing out the mistake. We have revised the mistakes in our text (line 46).
In few cases it may be caused by gonococcus, Chlamydia, herpes simplex virus, etc
What the author think about thisɁ
And what about the Pregnancy and other risk factor ….
Ans: Thank you for the comment. Because our previous studies have focused on the field of UPEC infection within bladder and prostate cells by regulating TLR-4 expression and JAK/STAT1 pathways, therefore in this study, we hope to continue to focus on the effect of insulin on UPEC-infected cells in a high-glucose environment. However, we believe that will be very interesting if we use our cell model on other pathogen infection. Maybe the JAK/STAT pathway also regulated the other infection beside UPEC. Actually we have started to test the parasite infection such as trichomonas in bladder cells, and hope to find some new insight.
Most cases of cystitis are caused by uropathogenic Escherichia coli (UPEC). Provide proper citation to this
Ans: Thank you for the comment. We have added the new reference in our text (reference 3).
Some studies can be precisely written as ‘a couple of studies’
Ans: Thank you for the comment. We have revised the description in our text.
Materials and methods, results and discussions are well written.
Ans: We thank the reviewer’s appreciation of our manuscript.

Round 2
Reviewer 1 Report
The authors present a much improved manuscript. However, there are a number of errors that detract from the significance of the publication. The authors should work to clarify the following concerns:
Major:
Figure 1. While the authors present improved data that can qualitatively support their conclusions, these plates still have too many bacterial colonies too count to accurately calculate a quantitative measure, especially with the 15 mM glucose and the 40 ul insulin. It would greatly strengthen the paper if the authors could further dilute (1:1000 or 1:10,000) and assess the CFUs. Once the authors count the number of colonies on the plate they can multiple that by the dilution factor (1:1000 or 1:10,000) to get an accurate CFU count. As of now significance values cannot be accurately calculated as plate counting is limited to 300 colonies per plate. More than 300 colonies are not able to be accurately counted.
Minor:
Abstract:
Lines 18-19: the authors clearly explain in their response that diabetic individuals have higher rates of UTI than non-diabetic individuals, but they do not correct the text. They should make these edits for clarity to the reader.
Introduction:
Line 48: the statement that “female genitals introduce cystitis” is not wholly true. Females are thought to have higher risk of UTI due to the anatomical architecture of their urogenital tract. The authors should fix this statement
Line 59: E. coli do not encode cilia. Only mammalian cells encode cilia. Do the authors mean pili?
Line 104: there appears to be a word or two missing? The cell lines were “grown/cultured in”? F-12K medium
Line 108: For clarity the authors should include “bladder” or “SV-HUC-1” to orient the reader. i.e. 5.5X10^5 bladder cells were incubated…
Line 109: the “e” is missing in the word “the”
Line 114: the “bacteria’s” invasion ability “was” detected, bacteria is missing the “s” and were should be “was”
Line 122: “For measure” should read “To measure”
Line 224: Figure 5 is discussed before figure 4. The figures should be renumbered or discussed in order
Line 361-362: the authors state that insulin reduces UPEC infection by limiting JAK/STAT signaling. While their data suggests this is true in order to confirm that the authors need to perform additional experiments where they prevent that signaling from occurring and test whether UPEC infection is then reduced. The authors should temper their language to reflect that.
The way the authors denoted “C”, “U”, “G”, “10”, “20”, and “40” in the figure legend of figure 6 is incredibly helpful. The authors should include these in each figure legend, or in the first figure legend and then again when additional controls are added in subsequent figures.
Author Response
11 17, 2021
Dear Editor:
Attached please find our revision manuscript entitled “Insulin downregulated the infection of uropathogenic Escherichia coli (UPEC) in bladder cells in a high-glucose environment through JAK/STAT signaling pathway “and the Manuscript Number of our paper is microorganisms-1417429.
We greatly appreciate all the valuable comments that have offered. These comments have helped us in improving the quality of the manuscript. The revisions are listed below for review, the changes we have made are highlighted with BOLD word in the text and their locations in the manuscript are indicated in response of each problem. Please review the updated manuscript and kindly let us know if additional changes are necessary. We sincerely hope that the revised manuscript and this letter will furnish sufficient explanation for the reviewer’s questions.
Sincerely yours,
Po-Ching Cheng, Ph. D.
Department of Molecular Parasitology and Tropical Diseases,
School of Medicine, College of Medicine,
Taipei Medical University,
Taipei, Taiwan
Reviewer1:
The authors present a much improved manuscript. However, there are a number of errors that detract from the significance of the publication. The authors should work to clarify the following concerns:
Ans: Thank you for these comments which helped to greatly improve our article, and we sincerely appreciate it.
Major:
Figure 1. While the authors present improved data that can qualitatively support their conclusions, these plates still have too many bacterial colonies too count to accurately calculate a quantitative measure, especially with the 15 mM glucose and the 40 ul insulin. It would greatly strengthen the paper if the authors could further dilute (1:1000 or 1:10,000) and assess the CFUs. Once the authors count the number of colonies on the plate they can multiple that by the dilution factor (1:1000 or 1:10,000) to get an accurate CFU count. As of now significance values cannot be accurately calculated as plate counting is limited to 300 colonies per plate. More than 300 colonies are not able to be accurately counted.
Ans: Thanks for reviewer’s comment. We have replaced the original figure 1 as a new one that limited the colonies < 300 in the plates by diluted to 1:10000. Related description also revised in the text (line 190-193, Figure 1 and figure legend 1).
Minor:
Abstract:
Lines 18-19: the authors clearly explain in their response that diabetic individuals have higher rates of UTI than non-diabetic individuals, but they do not correct the text. They should make these edits for clarity to the reader.
Ans: Thanks for pointing out the mistake. We have revised the description in the text to avoid misunderstand (line 18-19).
Introduction:
Line 48: the statement that “female genitals introduce cystitis” is not wholly true. Females are thought to have higher risk of UTI due to the anatomical architecture of their urogenital tract. The authors should fix this statement
Ans: Thanks for pointing out the mistake. We have revised the description in the text to avoid misunderstand (line 48-49).
Line 59: E. coli do not encode cilia. Only mammalian cells encode cilia. Do the authors mean pili?
Ans: Thanks for pointing out the mistake. We have revised the error in the text (line 58).
Line 104: there appears to be a word or two missing? The cell lines were “grown/cultured in”? F-12K medium
Ans: Thanks for pointing out the mistake. We have revised the error in the text (line 104).
Line 108: For clarity the authors should include “bladder” or “SV-HUC-1” to orient the reader. i.e. 5.5X10^5 bladder cells were incubated…
Ans: Thanks for pointing out the mistake. We have included “SV-HUC-1” in the text to avoid confusing (line 108).
Line 109: the “e” is missing in the word “the”
Ans: Thanks for pointing out the mistake. We have revised the error in the text (line 109).
Line 114: the “bacteria’s” invasion ability “was” detected, bacteria is missing the “s” and were should be “was”
Ans: Thanks for pointing out the mistake. We have revised the error in the text (line 114-115).
Line 122: “For measure” should read “To measure”
Ans: Thanks for pointing out the mistake. We have revised the error in the text (line 122).
Line 224: Figure 5 is discussed before figure 4. The figures should be renumbered or discussed in order
Ans: Thank you for pointing out the mistake. We have revised the description in the 3.2 section to improve the error (lines 228-236).
Line 361-362: the authors state that insulin reduces UPEC infection by limiting JAK/STAT signaling. While their data suggests this is true in order to confirm that the authors need to perform additional experiments where they prevent that signaling from occurring and test whether UPEC infection is then reduced. The authors should temper their language to reflect that.
Ans: Thank you for pointing out the mistake. We have revised the description in the discussion section to improve confusing (lines 370-374).
The way the authors denoted “C”, “U”, “G”, “10”, “20”, and “40” in the figure legend of figure 6 is incredibly helpful. The authors should include these in each figure legend, or in the first figure legend and then again when additional controls are added in subsequent figures.
Ans: Thanks for these comments. We have added the related descriptions in each figure legend (figure legend 1-5).
